# Heat-Bath and Metropolis Dynamics in Ising-like Models on Directed Regular Random Graphs

**DOI:** 10.3390/e25121615

**Published:** 2023-12-02

**Authors:** Adam Lipowski, António L. Ferreira, Dorota Lipowska

**Affiliations:** 1Faculty of Physics, Adam Mickiewicz University in Poznań, 61-614 Poznań, Poland; 2Departamento de Física, I3N, Universidade de Aveiro, 3810-193 Aveiro, Portugal; alf@ua.pt; 3Faculty of Modern Languages and Literatures, Adam Mickiewicz University in Poznań, 61-874 Poznań, Poland; lipowska@amu.edu.pl

**Keywords:** Ising model, directed random graphs, mean-field approximation, nonequilibrium systems

## Abstract

Using a single-site mean-field approximation (MFA) and Monte Carlo simulations, we examine Ising-like models on directed regular random graphs. The models are directed-network implementations of the Ising model, Ising model with absorbing states, and majority voter models. When these nonequilibrium models are driven by the heat-bath dynamics, their stationary characteristics, such as magnetization, are correctly reproduced by MFA as confirmed by Monte Carlo simulations. It turns out that MFA reproduces the same result as the generating functional analysis that is expected to provide the exact description of such models. We argue that on directed regular random graphs, the neighbors of a given vertex are typically uncorrelated, and that is why MFA for models with heat-bath dynamics provides their exact description. For models with Metropolis dynamics, certain additional correlations become relevant, and MFA, which neglects these correlations, is less accurate. Models with heat-bath dynamics undergo continuous phase transition, and at the critical point, the power-law time decay of the order parameter exhibits the behavior of the Ising mean-field universality class. Analogous phase transitions for models with Metropolis dynamics are discontinuous.

## 1. Introduction

The formulation of a number of statistical mechanics models was inspired by social dynamics [1]. Indeed, the dynamics of opinion formation [2], of epidemic spreading [3], or diffusion of innovations [4] can be, to some extent, described by a collection of interacting agents, whose states are represented by certain discrete, very often binary, variables. Such an approach bears some similarity to statistical mechanics phenomenology with an Ising model being a prime example [5]. Because statistical mechanics models originally intended to explain some thermodynamic properties of matter, their dynamics were constructed so as to reproduce the canonical equilibrium probability distributions [6]. Having in mind social dynamics applications, we are no longer obliged to impose such restrictions. This is important when modeling, for example, epidemic or rumor spreading since the corresponding models typically contain the so-called absorbing states and are thus much different from equilibrium systems [7].

Although statistical mechanics models are often formulated on regular lattices, such as a square lattice or a linear chain, more heterogeneous networks are usually considered in the context of social dynamics. Corresponding structures are often characterized with broad distributions of vertex degree [8], but also with temporal variability [9], adaptability [10], multilayered structure [11], or directedness of links [12]. Statistical mechanics models placed on such networks constitute a considerable challenge, and sophisticated mathematical methods must be used to examine their properties [13,14]. An important insight into the behavior of such models is also provided by various approximate methods, which very often are some kind of mean-field approximations (MFAs). A systematic approach to derive such approximations can be based on a master equation, as was demonstrated for several models with binary dynamics [15].

Of course, it is desirable to know the accuracy of such approximate methods. The experience that we have with the Ising model suggests that MFA on a complete graph, where each agent interacts with every other agent, should be exact [16]. On the other hand, interaction networks are usually less dense, which probably affects the accuracy of MFA. Perhaps, however, this is not always the case. Sometime ago, it was shown that for the Ising model on directed regular random graphs, MFA agrees with Monte Carlo simulations within a relative error ∼$10^{-5}$ [17]. In the present paper, we show that the accurate description within MFA is provided also for some other models on directed regular random graphs, namely, for the Ising model with absorbing states and the majority voter model. Moreover, we note the equivalence of MFA and the solution given within a generating functional method [18,19] for the Ising model and provide some arguments why, for models on directed regular random graphs and driven with heat-bath dynamics, MFA should describe these models exactly. We show that such a behavior results from the treelike structure of random graphs that effectively decouple the neighbors of a given site [13].

In some cases, our models undergo continuous phase transitions, and we also examine the time decay of the order parameter. For models with Metropolis-like dynamics, MFA turns out to be less accurate, which we relate to certain additional correlations generated by such dynamics, but apparently neglected by MFA. For the examined values of parameters, phase transitions in models with Metropolis dynamics turn out to be discontinuous.

## 2. Models

We examine models where binary variables $s_{i}=\pm 1$ are placed on the vertices (*i*) of a directed regular random graph of size *N*. Such graphs are generated with a straightforward algorithm, which randomly selects *z* neighbors (i.e., out-links) for each vertex (excluding connections to itself and multiple connections). As a result, we obtain the directed random graph where each vertex has *z* out-links. The number of in-links at a vertex has a Poisson distribution with the average value *z*. The evolution rules of spin variables mimic those often implemented in Ising ferromagnets, namely the heat-bath and Metropolis dynamics [6], suitably modified to encompass three classes of models—the Ising model, the Ising model with absorbing states, and the majority voter model. We performed Monte Carlo simulations of the above models and confronted the results with a single-site mean-field approximation (MFA).

Let us emphasize, however, that despite some similarity, like binary variables and up–down symmetry, such models are much different from an equilibrium Ising model, where a certain Hamiltonian determines the probability of spin configurations. Indeed, on directed graphs, detailed balance is, in general, broken, and such systems should be considered as nonequilibrium systems [20,21].

Ising-like models on directed graphs and with heat-bath dynamics were already examined using generating functional analysis [18,19] or a cavity method [22,23]. Within such techniques, one examines the exact equations governing the evolution of the order parameter. These techniques provide a correct description of such models and might be even used for graphs of arbitrary distribution. Let us notice, however, that in the heat-bath dynamics, we specify the probability to set a given spin in a given state. This is not the case in the Metropolis dynamics, where we specify the probability to flip a given spin. It is, in our opinion, not clear whether the above-mentioned powerful techniques might be used for models with Metropolis dynamics. Our studies are restricted to regular directed random graphs, and we demonstrate that for models with heat-bath and Metropolis-type dynamics, the behavior of these models is much different.

## 3. Results

### 3.1. Ising Model

#### 3.1.1. Heat-Bath Dynamics

First, we examine an Ising model with the heat-bath dynamics [6]. In such a case, after randomly selecting the spin, we determine the probability $P_{hb}$ to set the randomly chosen spin *i* as $s_{i}=1$. By analogy with equilibrium systems, $P_{hb}$ is given as
(1)$$P_{hb}=\frac{1}{1+\exp\left(-2h_{i}/T\right)}\;,\;\;\;\mathrm{where}\;h_{i}=\sum_{j_{i}}s_{j_{i}}$$
where *T* is a temperature-like parameter and $j_{i}$ are the (out-)neighbors of the site *i*, i.e., those *z* sites the site *i* is linked with. With the probability $1-P_{hb}$, the *i*-th spin is set to −1.

In our simulations, the initial configuration for each temperature was fully ferromagnetic ($s_{i}=1$). We measured the magnetization $m=1/N\sum_{i}s_{i}$, typically during $t_{MC}=10^{5}$ steps, where a step is defined as a single, on average, update of each site. In some calculations, where we analyzed the size dependence of our results, we used averages over 10 independent samples (generating a new graph in each sample). Before measurements, the model relaxed for $t_{MC}$ steps (usually, such a long relaxation was not needed because most of our simulations were outside the transition points).

Numerical results for z=8 are presented in Figure 1. They indicate that, in this model, the ferromagnetic phase (m>0) is replaced by the paramagnetic one (m=0) around T=7.

The behavior of such a model can be also examined using a single-site mean-field approximation. Namely, assuming that the (out-)neighbors of a given site are uncorrelated, we expect that, in the stationary state, the probability $P_{+}$ that a randomly chosen spin equals 1 satisfies the following equation:(2)$$P_{+}=\sum_{k=0}^{z}\frac{R_{z,k}}{1+\exp\left[-4(k-z/2)/T\right]}$$
where Rz,k=zkP+k(1−P+)z−k. In Equation (Equation 2), we assume some homogeneity, namely, that the probability for the neighbors of the chosen site to be equal 1 is also $P_{+}$. The term $R_{z,k}$ is the probability that, among *z* (out-)neighbors of a certain spin, there are *k* neighbors that are 1 and z−k that are −1. The denumerator in Equation (Equation 2) comes from the heat-bath dynamics probability of updating such a spin. The nonlinear Equation (Equation 2) can be easily solved numerically, and for z=8, the magnetization $m=2P_{+}-1$ corresponding to $P_{+}$ is plotted in Figure 1. One can notice that both MFA and numerical simulations are in perfect agreement. In particular, calculations for T=6.5 and for *N* up to $10^{6}$ (inset in Figure 1) show that MFA agrees with Monte Carlo simulations within an accuracy ∼$10^{-5}$.

The reason for such a good accuracy is not accidental. Indeed, it turns out for our model that MFA reproduces the exact solution of the model as obtained using generating functional analysis. In particular, the MFA description of the steady state as given by Equation (Equation 2) is exactly the same as the one obtained within the generating functional analysis [19]. The only difference is that the models examined within generating functional analysis implement the parallel version of the heat-bath dynamics, but the asynchronous version that we are using apparently leads to the same result.

Let us notice that, writing Equation (Equation 2), we assume that (out-)neighbors of a given site are uncorrelated. One can argue that for directed random graphs, such an assumption is likely to be satisfied. First, let us observe that neighboring spins $s_{a}$ and $s_{b}$ (Figure 2A) are influencing (correlating) a given spin $s_{c}$ (i.e., they enter the *h*-term in Equation (Equation 1)), but not vice versa. Namely, $s_{c}$ does not influence $s_{a}$ or $s_{b}$. To have $s_{a}$ and $s_{b}$ correlated, we thus need a direct link between them. However, for finite-*z* graphs, such links exist with a small probability ∼1/N. It means that for large graphs, most sites have uncorrelated neighbors, and that justifies the mean-field approximation (Equation (Equation 2)). Our argumentation on the validity of Equation (Equation 2) is only qualitative. For example, $s_{a}$ and $s_{b}$ could be correlated when we have a directed path of links instead of a single link, but short paths of this kind are unlikely on random graphs [13]. Actually, such a property implies that random graphs are locally treelike graphs. A more quantitative approach justifying Equation (Equation 2) would be certainly desirable, but the agreement with generating a functional method supports our considerations.

Let us also notice that on an undirected network (Figure 2B), the spins $s_{a}$ and $s_{b}$ are influenced by $s_{c}$, which makes them correlated (no matter whether they are connected by a direct link or not). In such a case, we expect that Equation (Equation 2) will be erroneous. For the Ising model on random regular undirected graphs, the critical temperature is known to be equal to $T_{c}=2/\ln\left[(z+1)/(z-1)\right]$ [24], and for z=8, we obtain $T_{c}=7.958\ldots$. This is different from the MFA Equation (Equation 2), which in this case predicts $T_{c}=7.062\ldots$ (as also seen in Figure 1). Actually, spin models on undirected random graphs, and also on certain more general treelike lattices, might be analyzed exactly using several techniques, such as the recursion method [24], the replica technique [25], or the cavity method [26].

One can generalize Equation (Equation 2) for Ising models on directed Erdös–Rényi random graphs, where it also shows a very good agreement with numerical simulations [21]. For z=2, 3, and 4, Equation (Equation 2) can be easily solved analytically [17].

Since our model exhibits a continuous phase transition, it might be of interest to examine some of its dynamical features at criticality. We analyzed the time decay of the order parameter m(t). Asymptotic decay of m(t) was already analyzed for a number of models [27], and it is expected that in the long-time regime at the critical point, one has m(t)∼$t^{-\beta /\nu \omega }$, where β and ν are exponents that describe the critical behavior of the order parameter and correlation length and ω are the dynamical critical exponents [28,29]. One might expect that our model belongs to the mean-field Ising universality, but taking into account its nonequilibrium features, such a behavior is by no means obvious, especially with respect to its dynamical behavior. Using the well-known mean-field values β=1/2, ν=1/2, and ω=2 [30] in the long-time limit, we obtain m(t)∼$t^{-1/2}$. To verify such an expectation, we calculated the average behavior of the time-dependent magnetization m(t) for z=8 and several temperatures. We made simulations for $N=10^{6}$, and the final results, which are averages over 100 independent samples, are presented in Figure 3. Our simulations demonstrate that at the critical point (T = 7.06), the decay of magnetization is consistent with the decay ∼$t^{-1/2}$. It shows that with respect to the dynamical characteristics, the behavior of our model is consistent with the mean-field Ising universality class.

#### 3.1.2. Metropolis Dynamics

The so-called Metropolis dynamics is yet another algorithm to simulate Ising models [6]. In this dynamics, one randomly selects a spin ($s_{i}$) and flips it with the probability min[1,exp(−ΔE/T)], where ΔE is the energy change experienced during such a flip. On directed graphs, by analogy with undirected graphs, we can define the (pseudo-)energy change as $\Delta E=2s_{i}h_{i}$, where $h_{i}$ is defined in Equation (Equation 1). With such an algorithm, we made Monte Carlo simulations, and the results for z=8 are presented in Figure 4.

The rules of the heat-bath and Metropolis dynamics were devised in such a way that when they drive an Ising model on undirected graphs, they both reproduce the same equilibrium probability distribution. On directed graphs, these dynamics do not obey a detailed balance, and they are unlikely to be equivalent. Indeed, as we can notice in Figure 4, the magnetization *m* vanishes at a substantially lower temperature than for the model with the heat-bath dynamics. What is more, with the Metropolis dynamics, the magnetization seems to have a discontinuous jump. It would be desirable to gain some understanding of such a behavior that is in qualitative contrast with continuous phase transitions that are more typical of Ising-like models. Let us notice that some other fine dynamical details are also known to induce a similar discontinuous behavior [31,32].

For the Ising model with Metropolis dynamics, we can also develop the mean-field approximation. In the stationary state, as described using MFA, we expect that the probability that a selected spin is 1 and it flips to −1 should be equal to the probability that it is −1 and it flips to 1. Using the Metropolis transition rates, we can thus write the following relation:(3)$$P_{+}\left[\sum_{k=0}^{z/2}R_{z,k}+\sum_{k=z/2+1}^{z}R_{z,k}\exp(2z-4k)/T\right]=\left(1-P_{+}\right)\left[\sum_{k=0}^{z/2-1}R_{z,k}\exp(4k-2z)/T+\sum_{k=z/2}^{z}R_{z,k}\right],$$
where, for simplicity, we assumed that *z* is an even number and $P_{+}$ is interpreted as a probability that a randomly chosen spin is 1.

Standard numerical methods can be used to solve the nonlinear Equation (Equation 3), and the solution for z=8 is shown in Figure 4. Good agreement of MFA with simulations can be seen but only at low temperature. Similar to Monte Carlo simulations, MFA also predicts a discontinuous jump of the magnetization at the transition temperature.

Clearly, MFA for the Ising model with the Metropolis dynamics is not very accurate. Of course, writing Equation (Equation 3), we also assumed that the neighbors of a chosen site are uncorrelated, but as we already argued, this is a plausible assumption. Let us notice, however, that in Equation (Equation 3), we are making yet another assumption, namely, that the considered spin is independent of the neighboring spins. In our opinion, this is the main source of the discrepancy between MFA and numerical simulations. In a more adequate treatment, we should introduce additional parameters describing such correlations and additional equations like Equation (Equation 3) that would enable their determination. Such an approach, often called pair approximation, as well as the single-site MFA analyzed in our paper, can be obtained in a more systematic way from a certain master equation, which describes a large class of stochastic models with binary dynamics [15].

We will not pursue such more elaborate mean-field approximations. Instead, we will confront numerical simulations with MFA in some other models with the heat-bath and Metropolis dynamics. Let us also notice that the generating functional analysis [18,19] is based on transition rates that are used in heath-bath dynamics, and it is not clear whether this technique might be extended for Metropolis dynamics.

### 3.2. Ising Model with Absorbing States

#### 3.2.1. Heat-Bath Dynamics

In a certain class of models, the rules of the Ising dynamics are modified so as to obtain models with absorbing states. Such systems violate a detailed balance, but they can exhibit a rich behavior with some nonequilibrium phase transitions [33,34]. They often retain an Ising up–down symmetry, which, combined with the absorbing-state transition, results in a voter-type critical point. Sometime ago, it was noticed that the voter critical point can split, and the breaking of the up–down symmetry and the collapse into an absorbing state would take place separately [35]. Such an effect was explained in terms of the Langevin description [36] and observed in some other systems [37,38,39,40].

The Ising model on directed networks retains the up–down symmetry, and to have the dynamics with absorbing states, we should modify the heat-bath dynamics so that the spin that has *z* neighbors in the same state is set in the same state (as neighbors) with the probability $P_{hb}=1$. It means that in the definition (Equation 1), we should set $P_{hb}=1$ for $h_{i}=z$ and $P_{hb}=0$ for $h_{i}=-z$. For the remaining configurations of neighbors, we keep the definition (Equation 1). For a model with such dynamics, the MFA equation takes the form
(4)$$P_{+}=R_{z,z}+\sum_{k=1}^{z-1}\frac{R_{z,k}}{1+\exp\left[-4(k-z/2)/T\right]}.$$

A numerical solution of Equation (Equation 4) and results from Monte Carlo simulations for z=8 are shown in Figure 5. Since the model dynamics contains some absorbing states (with all spins +1 or all spins −1), we start simulations from 80% of spins set as +1 and 20% as −1. At high temperature, the model remains in the disordered paramagnetic phase. At around T=5.33, the up–down symmetry gets broken, and the model is either positively (m>0) or negatively (m<0) magnetized. Upon further cooling, at T=4.97, the second transition takes place, and the model enters an absorbing state. As we already mentioned, a similar behavior has already been reported for the absorbing-state Ising model on some regular lattices (square lattice with further neighbor interactions or simple cubic lattice) [35]. Let us notice (inset in Figure 5) that MFA (Equation (Equation 4)), similar to the Ising model with the heat-bath dynamics, also provides an extremely accurate and possibly exact description of the model. Of course, Equation (Equation 4) is also based on the independence of neighboring spins, but as we argued in the previous subsection, for random directed regular graphs, such an assumption should be satisfied.

We do not present numerical results, but our analysis shows that for z=4, the intermediate phase (with 0<m<1) does not exist, and the model transitions from the paramagnetic phase directly to the absorbing state. In such a case, the symmetry breaking and the absorbing-state phase transition take place simultaneously, as in the voter model [41]. A similar behavior is observed for the absorbing-state Ising model on a square lattice, where with the nearest-neighbor interactions (z=4), both transitions take place at the same temperature, while the addition of further range interactions leads to their separation [35]. For z=2, our model remains in the paramagnetic phase for any T>0, similar to the Ising model (without absorbing states) on a directed regular random graph with z=2 [21] and, of course, to the one-dimensional (equilibrium) Ising model.

For the Ising model with absorbing states, we also examined the time decay of magnetization at the symmetry breaking transition. Our results (Figure 6) demonstrate that, also in this case, the decay ($∼t^{-1/2}$) is consistent with the Ising mean-field universality class.

#### 3.2.2. Metropolis Dynamics

We also analyzed the Ising model with absorbing states and the Metropolis update. To have the absorbing states, we suppress flips that would result in a maximum increase in energy (ΔE=2z). With such dynamics, the MFA equation is similar to Equation (Equation 3) and has the form
(5)$$P_{+}\left[\sum_{k=0}^{z/2}R_{z,k}+\sum_{k=z/2+1}^{z-1}R_{z,k}\exp(2z-4k)/T\right]=\left(1-P_{+}\right)\left[\sum_{k=1}^{z/2}R_{z,k}\exp(4k-2z)/T+\sum_{k=z/2+1}^{z}R_{z,k}\right].$$

The numerical solution of Equation (Equation 5) and the results from Monte Carlo simulations for z=8 are presented in Figure 7. Both methods predict the intermediate phase (0<m<1) and separate the symmetry-breaking and absorbing-state transitions. Similar to the Ising model, the symmetry breaking transition is discontinuous, and MFA and Monte Carlo simulations are noticeably different.

### 3.3. Majority Voter Model

#### 3.3.1. Heat-Bath Dynamics

The majority voter model is a frequently studied model of opinion formation [42,43,44]. In this model, the behavior of a given spin is determined by the sign (not the strength) of the majority of neighboring spins. In the heat-bath-like formulation of the majority voter model dynamics, we specify the probability $P_{hb}$ that a randomly chosen spin $s_{i}$ is set to 1 as
(6)$$P_{hb}=\left\{\begin{array}{cc} \frac{1+q}{2} & \;\mathrm{for}\;h_{i}>0\\ \;\;\frac{1}{2} & \;\mathrm{for}\;h_{i}=0\\ \frac{1-q}{2} & \;\mathrm{for}\;h_{i}<0 \end{array}\right.,$$
where $h_{i}$ is defined as in Equation (Equation 1) and the parameter *q* controls the level of noise.

For such a model, MFA leads to the following equation:(7)$$P_{+}=\frac{1}{2}\left(1+q\right)\;\sum_{k=0}^{z/2-1}\;R_{z,k}+\frac{1}{2}R_{z,z/2}+\frac{1}{2}\left(1-q\right)\;\sum_{k=z/2+1}^{z}\;R_{z,k}.$$

The numerical solution of Equation (Equation 7) and the results from Monte Carlo simulations for z=4 are shown in Figure 8. As expected, for large *q*, the model remains in the polarized phase (0<m<1), which around q=0.66(1) is replaced with the unpolarized phase (m=0). As in our other models with the heat-bath dynamics, as well as in this case, MFA is in perfect agreement with simulations, which is clearly demonstrated for more detailed calculations for q=0.75.

For the majority voter model, we also examined the time decay of magnetization. Our results (Figure 9) demonstrate that, also in this case, the decay ($∼t^{-1/2}$) is consistent with the Ising mean-field universality class.

#### 3.3.2. Metropolis Dynamics

One can also formulate the Metropolis version of the majority voter model. In particular, we introduce $P(s_{i}\to -s_{i})$ as the probability to flip the randomly chosen spin $s_{i}$ defined as follows:(8)$$P\left(s_{i}\to -s_{i}\right)=\left\{\begin{array}{cc} \;\;1 & \;\mathrm{for}\;h_{i}s_{i}\leq 0\\ (1-q)/2 & \;\mathrm{for}\;h_{i}s_{i}>0 \end{array}\right..$$ With such dynamics, the model can be analyzed within MFA, and the corresponding equation has the form
(9)$$P_{+}\left[\sum_{k=0}^{z/2}R_{z,k}+\frac{1}{2}\left(1-q\right)\sum_{k=z/2+1}^{z}R_{z,k}\right]=\left(1-P_{+}\right)\left[\frac{1}{2}\left(1-q\right)\sum_{k=0}^{z/2}R_{z,k}+\sum_{k=z/2+1}^{z}R_{z,k}\right].$$

The numerical solution of Equation (Equation 9) and the results from Monte Carlo simulations for z=4 are presented in Figure 10. A noticeable discrepancy of these results, especially in the vicinity of the transition, can be seen. Similar to other models with the Metropolis update, the transition between polarized and nonpolarized phases is most likely discontinuous.

## 4. Conclusions and Remarks

In the present paper, we examined Ising-like models on directed regular random graphs. We examined these models using mean-field approximation and Monte Carlo simulations. It turns out that when these models are driven by heat-bath dynamics, their steady-state behavior is correctly reproduced by MFA. Indeed, for an Ising model, our MFA is equivalent to the so-called generating functional analysis that is expected to provide the exact description of this model. Since, for the Ising model with absorbing states and the majority voter model, MFA also offers a very accurate description, as confirmed using Monte Carlo simulations, we expect that, in this case, MFA also provides the exact description. It would be interesting to apply generating functional analysis to these models to verify our expectations. It is perhaps worth emphasizing that MFA is a very simple approximation that, nevertheless, as our work demonstrates, in some cases, offers exact results. We gave some arguments that validate MFA for our models, namely, that on directed random graphs, the neighbors of a given spin are typically uncorrelated. We also examined some dynamical characteristics of our models with heat-bath dynamics. Numerical simulations show that, at criticality, magnetizations decay as $t^{-1/2}$, which confirms that although these models are nonequilibrium, their dynamics belong to the Ising mean-field universality class.

A much different behavior appears when these models are driven by Metropolis dynamics. Let us notice that, for equilibrium systems, both heat-bath and Metropolis dynamics are equivalent and drive the system toward the equilibrium state described by canonical distribution. For models on directed graphs, our models are nonequilibrium, and these two dynamics are not equivalent. For Metropolis dynamics, the independence of neighbors is not enough, and MFA turns out to be less accurate. The applicability of the generating functional analysis to models with Metropolis dynamics is, in our opinion, an open problem. It would be desirable to explain the nature of discontinuous transitions that we observed for the Metropolis dynamics both within MFA and in Monte Carlo simulations. Apparently, even a qualitative behavior of models on directed graphs, such as the nature of a phase transition, is sensitive to some details of dynamical rules.

It would be interesting to extend our work to some other dynamics, such as conservative Kawasaki dynamics [45]. One can also examine some other models defined with the heat-bath dynamics. For example, epidemic spreading models, like the SIR model, can be formulated in such a way, and we expect that on directed graphs, MFA should also provide their very accurate, perhaps exact, description. We also expect a similar efficiency of MFA for models with three (or more) state variables, e.g., the Ising model with a spin S=1 or some opinion formation models [44].

## Figures and Tables

**Figure 1 entropy-25-01615-f001:**
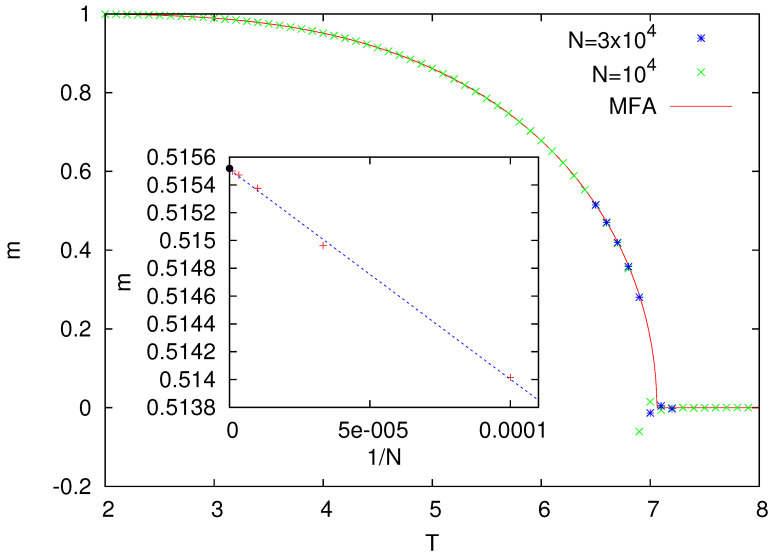
Magnetization *m* as a function of temperature *T* for the Ising model with the heat-bath dynamics (z = 8). The inset presents the size dependence of the magnetization as calculated for T=6.5. The black bullet shows the MFA value m=0.5155186…, as obtained from the numerical solution of Equation (Equation 2). From the linear fit, based on numerical data for $10^{4}\leq N\leq 10^{6}$, in the limit N→∞, we obtain 0.51551(1), which is extremely close to the MFA value.

**Figure 2 entropy-25-01615-f002:**
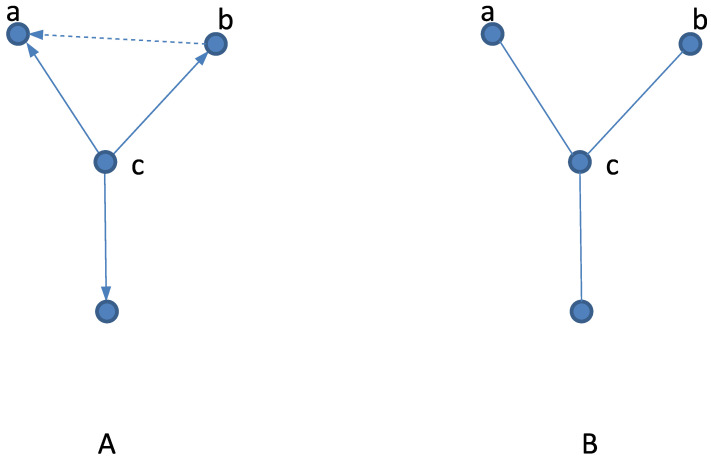
(**A**) On a directed network, (out-)neighboring spins $s_{a}$ and $s_{b}$ influence (correlate) $s_{c}$, but not vice versa. Namely, $s_{a}$ and $s_{b}$ are uncorrelated unless there is a direct link between them (dashed line). On a graph of size *N*, where each vertex is connected to *z* other vertices, such a link exists with a probability N−1z−1/N−1z=z/(N−z)∼1/N; i.e., on a large random directed graph, neighbors of a given site are mainly uncorrelated. (**B**) On an undirected network, neighboring spins $s_{a}$ and $s_{b}$ influence $s_{c}$, but also $s_{c}$ influences $s_{a}$ and $s_{b}$. Thus, on an undirected network, neighbors of a given spin $s_{c}$ are correlated (at least via $s_{c}$).

**Figure 3 entropy-25-01615-f003:**
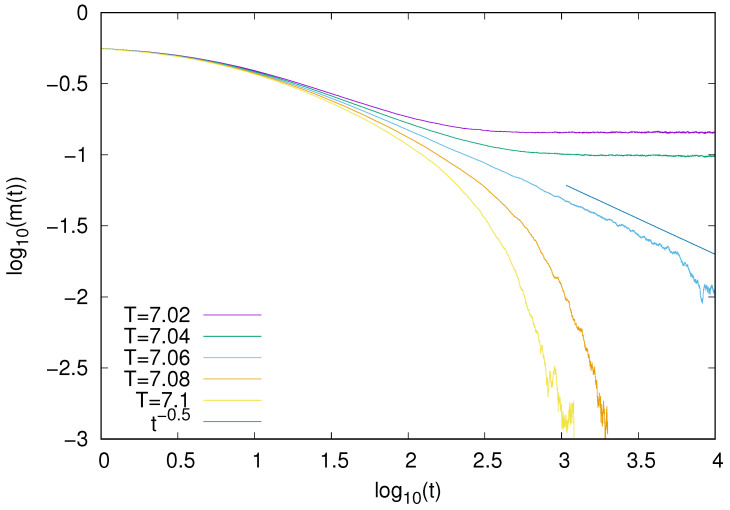
Time decay of the magnetization m(t) for the Ising model with the heat-bath dynamics (z=8). At the critical point (T = 7.06), magnetization seems to decay as $∼t^{-1/2}$.

**Figure 4 entropy-25-01615-f004:**
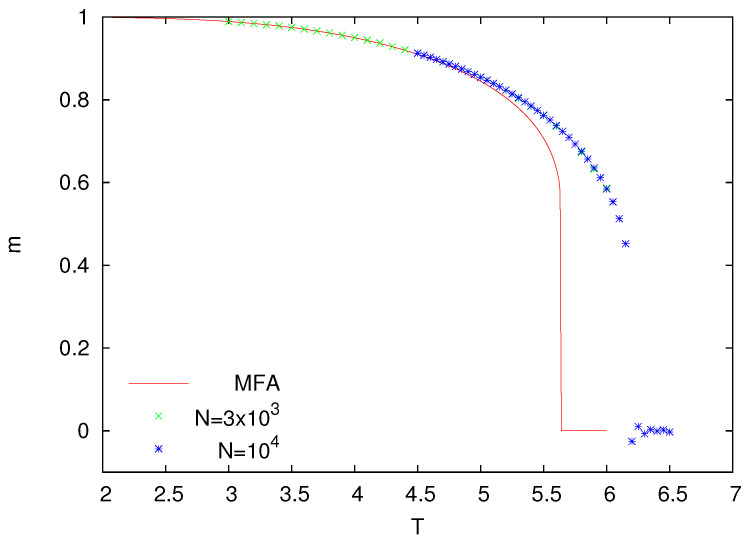
Magnetization *m* as a function of temperature *T* for the Ising model with Metropolis dynamics (z = 8). MFA denotes numerical solutions of Equation (Equation 3).

**Figure 5 entropy-25-01615-f005:**
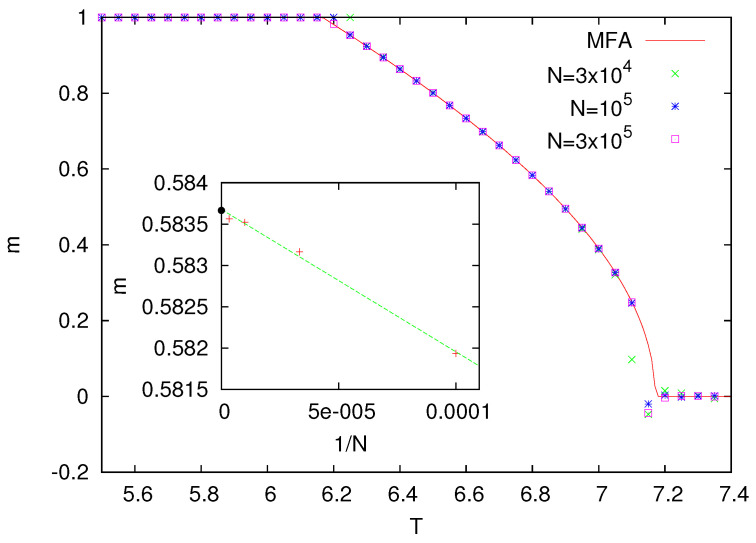
Magnetization *m* as a function of temperature *T* for the Ising model with absorbing states and with the heat-bath dynamics (z=8). The inset presents the size dependence of the magnetization as calculated for T=6.8. The black bullet shows the MFA value m=0.583665…, as obtained from the numerical solution of Equation (Equation 4). From the linear fit, based on numerical data for $10^{4}\leq N\leq 10^{6}$, in the limit N→∞, we obtain 0.5837(2), which is extremely close to the MFA value.

**Figure 6 entropy-25-01615-f006:**
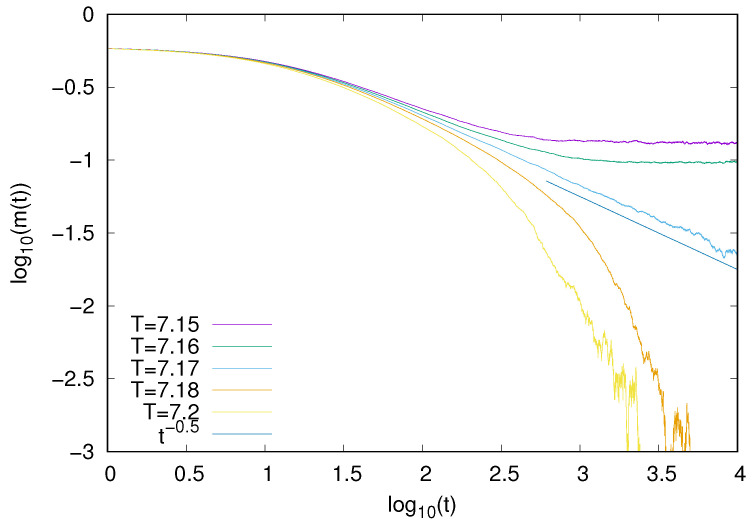
Time decay of the magnetization *m* for the Ising model with absorbing states with the heat-bath dynamics (z=8). At the critical point (T = 7.17), magnetization seems to decay as $∼t^{-1/2}$.

**Figure 7 entropy-25-01615-f007:**
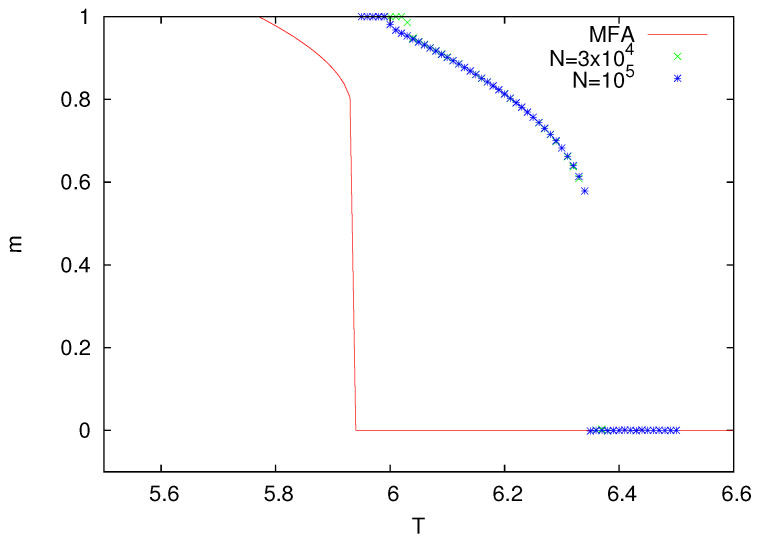
Magnetization *m* as a function of temperature *T* for the Ising model with absorbing states and the Metropolis dynamics (z=8). MFA denotes a numerical solution of Equation (Equation 5).

**Figure 8 entropy-25-01615-f008:**
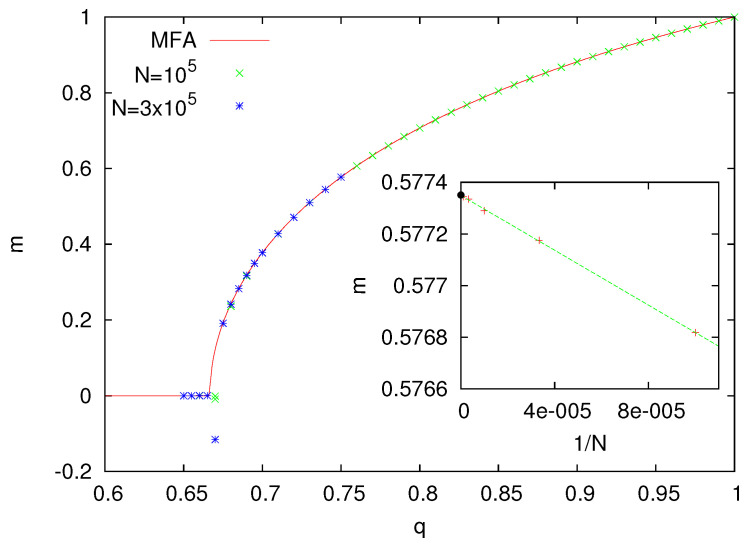
Magnetization *m* as a function of the parameter *q* for the majority voter model with the heat-bath dynamics (z=4). The inset presents the size dependence of the magnetization as calculated for q=0.75. The black bullet shows the MFA value m=0.577350…, as obtained from the numerical solution of Equation (Equation 7). From the linear fit, based on numerical data for $10^{4}\leq N\leq 10^{6}$, in the limit N→∞, we obtain 0.577349(3), which is extremely close to the MFA value.

**Figure 9 entropy-25-01615-f009:**
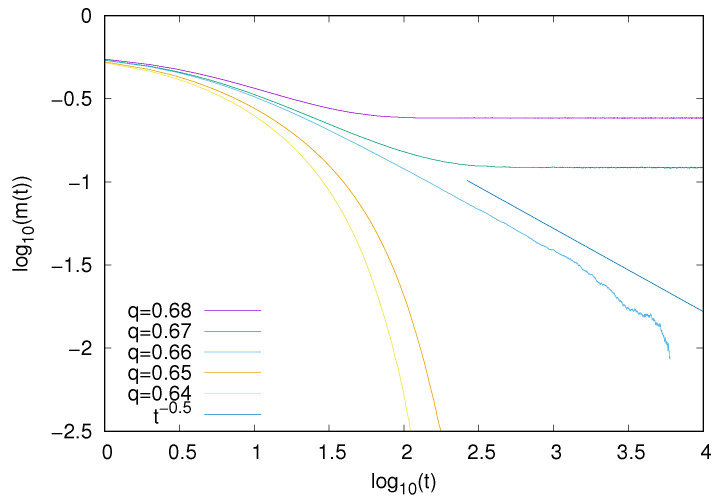
Time decay of magnetization *m* for the majority voter model with the heat-bath dynamics (*z* = 4). At the critical point (q = 0.66), magnetization seems to decay as ∼$t^{-1/2}$.

**Figure 10 entropy-25-01615-f010:**
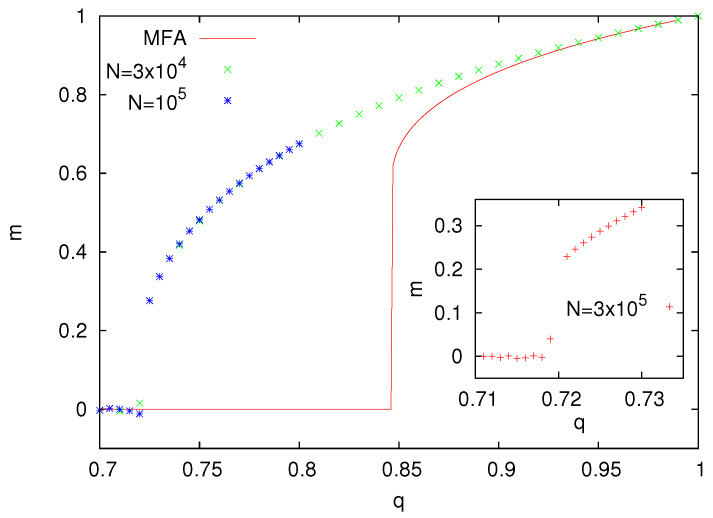
Magnetization *m* as a function of the parameter *q* for the majority voter model with the Metropolis dynamics (z=4). The plot of the magnetization in the vicinity of the transition (inset) suggests its discontinuous behavior.

## Data Availability

Data is contained within the article.

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
