# Peer review of "Heat-Bath and Metropolis Dynamics in Ising-like Models on Directed Regular Random Graphs"

_entropy, 2023, doi:10.3390/e25121615_

Round 1

Reviewer 1 Report

Comments and Suggestions for Authors

The authors have diligently addressed an important issue that I raised in my review, and I would like to extend my congratulations for their dedication to improving the manuscript. They have investigated magnetization decay at critical points in the context of short-time dynamics, which is both easily comprehensible and computationally efficient. I am confident that their efforts have significantly enhanced the quality of this manuscript, and I wholeheartedly recommend its publication.

Author Response

We thank the Reviewer for favourable opinion.

Reviewer 2 Report

Comments and Suggestions for Authors

It's a nice paper studying a set of natural problems on a very simple substrate - directed random regular graph. The paper is clearly written. The results are mostly to be expected. Indeed, the construction of the graph guarantees that (in the limit of large N) the immediate neighbours of a given node are uncorrelated, so (at least for models with well-defined equilibrium) the MFA must be exact. 

The minor suggestions I have are as follows:
1) the fact that the RRG is locally tree-like (with the minimal size of a cycle diverging logarithmically with N \to \infty) should be emphasized more clearly, I think it is good to discuss this in the introduction, as well as to explain why it makes MFA exact;

2) Lines 128 and below, when the undirected RRG is discussed. The authors are right that in this case there are correlations between neighbouring spins. But this correlations are in a sense trivial, they can be taken into account exactly by, e.g., cavity method (see Spin glass theory and beyond by Mezard, Parisi and Verasoro). In fact, that is exactly the reason why the formula in line 132 is exact. For locally non-tree-like lattices no such exact result is available. I think it is worth briefly discussing this topic.

3) Frankly, I am a bit surprised why MFA for the Metropolis model works so badly. The authors in lines 181-185 suggest a way of improving it. Is it possible to write down the corresponding equations and solve them numerically? If yes, and if it is shown that second (pair) approximation fits the simulations better, it will validate the authors' argument. If it is not too difficult to do, I would advise adding such a consideration to the text.

Summing up, I think the paper is reasonable and interesting, and can be published after authors consider optional suggestion outlined above.

Comments on the Quality of English Language

The English is the ok.

Author Response

We thank the reviewer for a number of constructive comments.

  • In Introduction (l. 49) we added the sentence “We show that such a behaviour results from the tree-like structure of random graphs that the effectively decouple the neighbours of a given site”. In the main text (l. 115-117) we modified the text into into:  For example $s_a$ and $s_b$ could be correlated when we have a directed path of links instead of a single link but short paths of this kind are on random graphs unlikely. Actually, such a property  implies that random graphs are locally tree-like graphs.
  • In Section 3.1.1. (l. 125-127) we mentioned that on undirected random graphs spin models might be analysed exactly using several techniques such as recursion method, the replica technique, or the cavity method. We also cited the paper by Leone et al [25], and the book by Mezard et al.[26].

  • We thank the referee for prompting us to use the pair approximation for models with Metropolis dynamics. Indeed, in the literature there are examples that show that this method improves the single-site approximation but the improvement is usually only quantitative. Actually, we did some calculations along this line and noticed that while for the Ising model (on directed RRG) the pair approximation considerably improves  the single site MFA, for other our models the improvement is rather small. In the near future we plan to analyse this approximation in some more details, possibly extending the method beyond the pair approximation.

Reviewer 3 Report

Comments and Suggestions for Authors

This manuscript is easy to read, and it is straight forward. It provides interesting results and points out interesting avenues of analysis that go beyond the scope of this work, which applies single site mean field theory and Monte Carlo simulations using different forms of spin dynamics. The interesting aspect of this work is the results they obtained on directed graphs. Even though the spin dynamics is taken from normal equilibrium statistical physics methods, the results depend on the form of dynamics (normally not the case for non-directed graphs) due to subtleties in how information is correlated (or lack thereof) within the directed graph. The directed graph puts the system out of equilibrium, and different results are obtained from the expected results that would otherwise arise in the equilibrium case. In my view the manuscript can be published as is. Furthermore, the introduction is sufficient as is, with a general background about the utility of the Ising-model on directed graphs that connect to social dynamic models. The conclusions are clear and not overstated, and there are nice suggestions about open problems.

Having said this, minor edits are still needed to fix English grammar. 

My last comment is that I have not worked in this area for some time, yet, I did not made an extensive literature review to verify that this work is novel as implied by the authors. Based on the authors writing, my sense is that this work is novel, and the authors make it clear what their contributions are. In particular, they motivate their work in terms of application of an Ising-model where the underlying connectivity is on a directed graph.   

Comments on the Quality of English Language

Minor edits on the grammar is needed. Even in the abstract, Ising should not be left on its own, but rather say Ising model. Some articles (like the) are missing here and there, and in general a careful proof read is required as there are many sentences that are slightly off in the way they are phrased. Despite this, I was able to understand what the authors wrote, and I do not need to re-read a revision because the scientific content is clear and interesting.

Author Response

We thank the Reviewer for favourable opinion.  We corrected abstract as suggested. We also read once more the paper and corrected some English mistakes.